# Factors Influencing Longevity of Humoral Response to SARS-CoV-2 Vaccination in Patients with End Stage Kidney Disease Receiving Renal Replacement Therapy

**DOI:** 10.3390/jcm11174984

**Published:** 2022-08-25

**Authors:** Irena Glowinska, Barbara Labij-Reduta, Jerzy Juzwiuk, Magdalena Lukaszewicz, Adam Pietruczuk, Agata Poplawska, Anna Daniluk-Jamro, Katarzyna Kakareko, Alicja Rydzewska-Rosolowska, Beata Naumnik, Ewa Koc-Zorawska, Marcin Zorawski, Tomasz Hryszko

**Affiliations:** 12nd Department of Nephrology and Hypertension with Dialysis Unit, Medical University of Bialystok, ul. M. Sklodowskiej-Curie 24A, 15-276 Bialystok, Poland; 21st Department of Nephrology and Transplantation with Dialysis Unit, Medical University of Bialystok, ul. Zurawia 14, 15-540 Bialystok, Poland; 3Dialysis Unit, SPZOZ in Hajnowka, ul. A. Dowgirda 9, 17-200 Hajnowka, Poland; 4Nephrology and Dialysis Unit, Regional Specialist Hospital in Suwalki, ul. Szpitalna 60, 16-400 Suwalki, Poland; 5Dialysis Unit, SPZOZ in Sokolka, ul. W. Sikorskiego 40, 16-100 Sokolka, Poland; 6Nephrology and Dialysis Unit, Regional Specialist Hospital in Lomza, Al. J. Piłsudskiego 11, 18-404 Lomza, Poland; 7Dialysis Unit, SPZOZ in Lapy, ul. J. Korczaka 23, 18-100 Lapy, Poland; 8Department of Clinical Medicine, Medical University of Bialystok, 15-254 Bialystok, Poland

**Keywords:** SARS-CoV-2, vaccine, dialysis, humoral response, obesity

## Abstract

COVID-19 has severely affected the population of patients with end stage renal disease. Current data have proved a two-dose vaccination schedule against SARS-CoV-2 to be effective among dialyzed patients. There are limited data on the longevity and modulating factors of humoral response after vaccination. We performed a prospective longitudinal cohort study to determine longevity of the humoral response after SARS-CoV-2 vaccine. The study included 191 adult patients on hemodialysis and peritoneal dialysis. All participants had been vaccinated with three doses, either with BNT162b2 (Pfizer-BioNTech) (*n* = 109) or mRNA-1273 (Moderna) (*n* = 82). Anti-spike protein receptor-binding domain antibodies (anti-S IgG) were assessed using SARS-CoV-2 (RBD) IgG ELISA EIA-6150 IVD assay at baseline, on the 21st day and 43rd day, before a booster dose and two weeks thereafter. We found that before vaccination, 37.7% of the cohort had anti-S IgG titres concordant with seroconversion. After two-dose vaccination, seroconversion occurred in 97% of patients. The booster dose evoked a ~12-fold increase in antibody level. Obesity increased more than two-fold the odds for a decrease in anti-S IgG. Previous COVID-19 infection enhanced longevity of the humoral response following vaccination. In patients with previous COVID-19 infection, the BNT162b2 vaccine was associated with a higher odds of anti-S IgG waning compared to the mRNA-1273 vaccine. In conclusion, we report that obesity predisposes patients to protective antibody waning, hybrid immunity enhances odds for higher anti-S IgG concentrations and vaccine efficacy may be influenced by previous SARS-CoV-2 infection. The results might provide a rationale for vaccination protocol design.

## 1. Introduction

Patients on kidney replacement therapy are at high risk for serious illness and death due to COVID-19. Hospitalization is required in up to 50% of cases, and almost one third of these patients die [1,2]. 

The main risk factors for severe COVID-19—such as advanced age, diabetes, and cardiovascular disease—are commonly seen in dialysis population [3]. Moreover, the logistical aspects of maintenance hemodialysis (transportation, changing rooms, etc.) increase the risk for disease transmission. The advent of SARS-CoV-2 vaccinations has provided hope for this vulnerable population, and immunization programs are ongoing in dialyzed patients all over the world. Data from recent studies have proved a two-dose schedule to be effective among dialysis patients, although it has been found to evoke a lower humoral response in comparison to healthy controls [4]. There are scarce data reporting on the longevity of humoral response after vaccination, suggesting the diminished sustainability of protective antibody titres [5]. It should be noted that reports show increased rates of COVID-19 breakthrough infections among individuals with lower antibody titres. This raises a question about the optimal vaccination and surveillance strategy among dialysis patients, which may differ from the general population, as has been shown regarding, e.g., hepatitis B virus vaccination. The importance of the issue is highlighted by data showing long-term health consequences following acute SARS-CoV-2 infection, referred to as post-COVID-19 condition or Long COVID. Exposing this vulnerable population to another risk factor for a cardiovascular event [6], respiratory dysfunction [7] or diabetes development [8], etc., endangers their lives. 

Understanding the response to vaccination, the sustainability of protective antibody levels, and the predictive factors associated with waning humoral immunity are crucial to guiding optimal vaccination. Thus, we performed a study of dialyzed patients in the north-eastern part of Poland to determine the frequency and level of antibodies against SARS-CoV-2 in vaccine-naïve hemodialysis patients, as well as their antibody response to two-dose vaccination and a booster dose. Associations between the humoral response to the vaccination and COVID-19 history, type of vaccine and other sociodemographic as well as laboratory data were evaluated. Additionally, the longevity of antibodies against SARS-CoV-2 over a 6-month period, and factors associated with humoral immunity waning were assessed. 

## 2. Materials and Methods

### 2.1. Study Design

It was a multi-center prospective cohort study, which included 191 adult patients on maintenance hemodialysis (*n* = 181) and on peritoneal dialysis (*n* = 10) from 7 outpatient dialysis centers in the north-eastern part of Poland. All participants had been vaccinated either with BNT162b2 (Pfizer-BioNTech) (*n* = 109) or m-RNA-1273 (Moderna) (*n* = 82) vaccines. The study was started in January 2021 and finished in January 2022. Participation in the study was proposed to all dialysis centers in the north-eastern part of Poland to avoid selection bias. Seven out of eight centers agreed to take part in the study. Enrollment in the study was proposed to every patient in participating centers. There were no exclusion criteria. Out of 432 patients, 191 provided informed consent for participation in the study.

### 2.2. Vaccination Protocol

All participants were vaccinated with BNT162b1 or m-RNA1273 vaccines. All patients but one received three doses of the same vaccine. This patient was switched from mRNA-1273 to BNT-162b2 booster. The second dose was administered 3 weeks after the initial vaccination. Patients received two doses of BNT162b2 or m-RNA-1273. The booster dose was given after 6 months (182 ± 2 days). It was proposed to all study participants regardless of the humoral response 6 months after administration of the second dose.

### 2.3. Anti-S-IgG Antibody Determination

Concentrations of anti-spike protein receptor-binding domain antibodies (anti-S IgG) were assessed at baseline; before the second dose (day 21); 3-weeks after administration of the 2nd dose (day 43); before a booster dose (6 months); and two weeks thereafter. Anti-S IgG antibodies were determined using SARS-CoV-2 (RBD) IgG ELISA EIA-6150 IVD assay (DRG Instruments GmbH, Marburg, Germany). According to the manufacturer’s instructions, a value of anti-S IgG antibodies > 15.0 DU/mL was considered as evidence of seroconversion. 

### 2.4. Definitions 

Obesity was defined as Body Mass Index (BMI) > 30 kg/m^2^ and was calculated using the patient’s dry weight in kilograms. Kt/V, a measure of dialysis adequacy, was calculated using Daugirdas’ formula: K—the amount of urea completely cleared from blood in ml/min, t—time in minutes, V—volume of urea distribution in milliliters. According to the manufacturer, values of anti-S IgG in the range of 25.02–30.59 (IU/mL) are considered borderline; therefore, before computations of predictive factors of an adequate humoral response, they were classified as a lack of seroconversion. Patients’ comorbidities were determined from medical records. COVID-19 was diagnosed based on clinical symptoms and a positive PCR test.

### 2.5. Ethical Issues 

The study was conducted in accordance with the Helsinki Declaration, and the protocol was approved by the Medical University of Bialystok’s ethics committee (APK.002.101.2021). All participants provided informed consent. 

### 2.6. Statistical Analysis 

Continuous variables are reported as medians, as well as the first and third quartiles (Q1–Q3). Categorical data are reported as absolute and relative frequencies. Continuous variables were compared with Student’s *t*-test or the Mann–Whitney U test. Differences in categorical data were assessed with X^2^ test or Fisher’s exact test depending on whether the assumptions were met. 

An assessment of predictive factors of an adequate humoral response was done based on patients’ records available at baseline (*n* = 191), before the 2nd dose (*n* = 191), and two weeks after administration of booster (*n* = 91). Patients who were lost to follow up were excluded from the analysis. 

Predictors of an anti-S IgG waning were assessed 6 months after the vaccination, and included data from 139 patients.

Logistic regression was used to search for predictive factors of an adequate humoral response after each vaccination, as well as antibody waning after 6 months. First, the prediction of the likelihood of patients having an adequate humoral response after each dose of vaccine or an antibody waning after 6 months was assessed with a univariable logistic regression. If variables tended to be associated with the antibody waning (*p* < 0.15) in univariable analysis, they were input into multivariable model to evaluate independent associations. Interactions were assessed between all independent variables included in the model and if significant were entered into the model. Data were reported as odds ratio and 95% confidence interval. A sensitivity analysis was performed after removal of patients who contracted COVID-19 during the study period to confirm the robustness of the findings. Results were considered statistically significant if the *p*-value was <0.05. All statistical analyses were carried out with R ver. 4.0.3 (R Core Team, Vienna, Austria) [9]. 

## 3. Results

### 3.1. Study Population 

The study flow chart is presented in Figure 1. 

Before the first vaccination, 72 patients (37.7%) of the cohort had anti-S IgG titres concordant with seroconversion and therefore were considered to have had a prior COVID-19 infection. Forty-seven patients (65.2%) out of seventy-two seropositive prior to vaccination had symptomatic SARS-CoV2 infection confirmed with a PCR test. The other 25 patients (34.7%) were asymptomatic so they did not perform a PCR test. PCR was the sole type of test used to confirm SARS-CoV-2 infection. During the study, 10 patients contracted COVID (1 patient after first dose, 6 pts after second dose and 3 pts after booster). Five patients (4%), who had previously had COVID-19, were seronegative before vaccination as their immunity waned.

The characteristics of the patients are presented in Table 1. 

During the study period, 100 patients were lost to follow-up due to: refusal of the 3rd dose (*n* = 32), death (*n* = 24), transplantation (*n* = 6), hospitalization (*n* = 8), transfer to another center (*n* = 10), or vaccination outside of the center (*n* = 20). 

### 3.2. Anti-SARS-CoV-2 Humoral Response following Vaccination

After the first dose, the median anti-S IgG level increased significantly from 1.70 IU/mL [0.66–30.01] to 41.25 IU/mL [4.73–833.97] on the 21st day. The lack of a borderline or adequate response was noted in 43 (22.5), 6 (3.1%) and 142 (74.3%) patients, respectively. 

The second dose of the vaccine was administered after 21 days. The median anti-S IgG level three weeks after the second dose increased to 785.70 IU/mL [260.60–1781.50]. The number of non-responders decreased to six (3.1%) patients. There were no patients with borderline responses, so in 185 (97%) patients, full seroconversion was observed.

After 6 months, the median anti-S IgG level decreased by 34% to 515.51 IU/mL [104.47–1936.14]. This increased the number of patients without seroconversion to 11 (7.9%), as well as patients with borderline anti-S Ig G titre to 4 (2.9%), and decreased the number of patients with seropositivity to 124 (89.2%). 

Administration of the booster dose caused an increase in the median anti-S IgG level to 5306.65 IU/mL [1935.37–10217.60], which provided an adequate anti-S IgG level in 89 patients (97.8%). Two patients did not respond to the third vaccine dose (2.2%). Those two patients had end stage kidney disease due to ANCA vasculitis and were not on immunosuppressive therapy at that time. Humoral response to the vaccination is shown in Figure 2. 

### 3.3. Predictive Factors of Anti-SARS-CoV-2 Humoral Response

The following factors were assessed regarding the predictive value of an adequate response after the first, second, and third doses of the vaccine: age, sex, obesity, kt/V, hemoglobin concentration, dialysis modality, type of vaccine, presence of diabetes mellitus, hypertension, heart failure, and atrial fibrillation. After the first dose, BNT162b2 vaccine and Kt/V predicted a protective anti-S IgG level. These associations disappeared after the second and third doses (Table 2).

Previous SARS-CoV-2 infection significantly enhanced vaccination efficacy after the first dose (1266.50 IU/mL [565.75–2286.00] vs. 7.87 IU/mL [1.89–39.9], *p* < 0.001), the second dose (1740.00 IU/mL [1181.25–4153.25] vs. 414.90 IU/mL [174.20–921.95], *p* < 0.001), but not after the third dose (6107.90 IU/mL [3174.46–10542.05] vs. 4414.06 IU/mL [1638.73–9785.00], *p* = 0.41), as shown in Figure 3.

### 3.4. Predictive Factors of Anti-S IgG Waning

In the following step, predictive factors of a decrease in anti-S IgG after 6 months were assessed. Individuals prone to a decrease in antibody level were obese, SARS-CoV-2 naive before vaccination and vaccinated with mRNA-1273, had a higher hemoglobin concentration and were likely to have diabetes (*p* = 0.10). 

All the above variables were tested for interactions. There was one significant interaction between COVID-19 infection prior to vaccination and a type of vaccine used (BNT162b2; β = 2.81, SE 1.02, z value 2.75, *p* = 0.01). Multivariable logistic regression was used to study the relationship between obesity, COVID-19 infection before vaccination, the type of vaccine, hemoglobin concentration, the presence of diabetes, the type of vaccine used for the vaccination of patients with prior COVID-19 infection, and the waning of anti-S IgG titers after 6 months. It was found that being obese rose the odds of a decrease in anti-S IgG level by 225%, and a history of COVID-19 infection before vaccination decreased the odds of a decline in anti-S IgG level by 64%. Vaccination of patients with COVID-19 infection history before vaccination with BNT162b2 predisposed them to ani-S IgG waning (Table 3, Figure 4). To confirm the robustness of the obtained results, a sensitivity analysis was performed after patients who had contracted COVID-19 during the study period were removed from the patient cohort. It yielded the same findings, except for the vaccine type, which was no longer an independent predictor of a decrease in anti-S IgG level (Table 4).

## 4. Discussion

In this study, we reported on the humoral response after the first, second, and booster doses of the SARS-CoV-2 vaccine in dialyzed patients, and analyzed the predictive factors of the patients’ response to vaccination. Before the first dose of the vaccine, 37.7% of the study participants had evidence of previous infection. This percentage is comparable to that reported previously in a similar population [10]. After vaccination with two doses, seroconversion occurred in 97% of patients. Reported previously seroconversion rates after the two-dose vaccination protocol varied from 70.5% [11] to 96% [12], which places our data in the upper range. It seems that vaccination efficacy is increased in groups with higher rates of previous SARS-CoV-2 exposure [10] and varies accordingly depending on the time interval between the administration of the two doses [13]. This might explain the high level of seroconversion in our population. Indeed, the most robust antibody response was observed among patients infected before the vaccination after the first (~160-fold higher) as well as the second dose (~4-fold higher). This observation is in line with previous reports in the general population [14] and in dialyzed patients [10]. 

The waning of humoral immunity against SARS-CoV-2 has been described in the general population as well as in dialyzed patients, irrespective of whether it was conferred after infection or vaccination [15,16]. In our study, six months after the second dose of vaccine, the median anti-S IgG level decreased by 34% and resulted in fewer patients with seroconversion. The booster dose evoked a ~12-fold increase in antibody level to a similar extent in previously infected as well as SARS-CoV-2 naïve patients. The concentration of antibodies reached the highest level here in the whole study period. These data confirm that the booster dose is effective and restores antibody levels in dialyzed patients. Moreover, it might be speculated that the booster dose provides immune protection in COVID-19 naïve patients to the same extent as is seen in hybrid immunity, e.g., in persons who were vaccinated after being infected with SARS-CoV-2. However, it must be stressed that, as was shown by Cho et al. [17], while boosting increases plasma neutralizing capacity, there are still concerns if the antibodies are equivalent to those which are seen in convalescent individuals. 

Although protective antibody waning is a well described phenomenon, there are scarce data regarding the factors involved in this process, especially in dialyzed patients [18]. Based on our data, we were able to establish factors associated with the decrease in anti-S IgG following SARS-CoV-2 vaccination. It was found that obesity increased more than two-fold the odds for a decrease in protective antibodies following vaccination with a two-dose schedule in dialyzed patients. To the best of our knowledge, this is the first report pointing to obesity as a risk factor for protective antibody waning among dialyzed patients. According to our data, obesity increased by almost four times the odds for antibody waning after half a year. Obese people are characterized by a decreased number and impaired function of B lymphocytes and the altered function of effector memory T cells. It is speculated that the above derangements are caused by leptin and insulin resistance. These alterations may hinder the sustainability of humoral protection induced by vaccination. Interestingly, we did not find a negative effect of obesity on vaccine efficacy, which is in line with a recent statement from the Obesity Society [19] and does not accord with previous reports, at least in terms of SARS-CoV-2 vaccination [20] in dialyzed patients. 

In contrast to obesity, previous COVID-19 infection enhanced the longevity of the humoral response following vaccination in dialyzed patients. Similar data were published recently on the general population [21]. It was shown that in convalescent individuals there is retainment and expansion of memory B cells in response to vaccination [22], which provides long-lasting immunity. It is suggested that these changes may even last for up to one year. Our findings enrich a previous report [15] regarding the efficacy of different types of anti-SARS-CoV-2 vaccines in dialyzed patients. We found that previous SARS-CoV-2 infection may alter the used vaccine’s efficacy regarding the longevity of anti-S IgG antibodies. Vaccination with BNT162b2 of patients with prior COVID-19 infection negatively influenced protective antibody longevity. Although the confidence interval was wide, the effect size should be classified as large. It should be noted that all described predictive factors of anti-S IgG waning held in the sensitivity analysis, which confirms their robustness. We are aware that our results do not allow speculation as to whether this translates into any clinically meaningful effect, as there are reports showing the superiority of the mRNA-1273 vaccine in relation to infection and SARS-CoV-2-related hospitalization [23]. 

Our data have clinical implications as they may provide a rationale for the selection of hemodialyzed patients who are prone to faster antibody waning and may therefore be offered a booster at a shorter time interval and with a specific type of vaccine. 

### Limitations

The high drop-out rate observed in our study hindered our ability to detect all factors associated with humoral response waning among hemodialyzed patients. Moreover, as clinical endpoints were not assessed, the interpretation of our results in a clinical context is limited. The observational nature of the study per se did not allow us to establish the causation of observed phenomena. 

## 5. Conclusions

In conclusion, we report for the first time (to the best of our knowledge) that obesity increases the odds of waning antibody levels, while hybrid immunity enhances the odds of higher anti-S IgG concentration following vaccination in dialyzed patients. The efficacy of a vaccine may depend on previous SARS-CoV-2 infection. These results might provide a rationale for vaccination protocol design. 

## Figures and Tables

**Figure 1 jcm-11-04984-f001:**
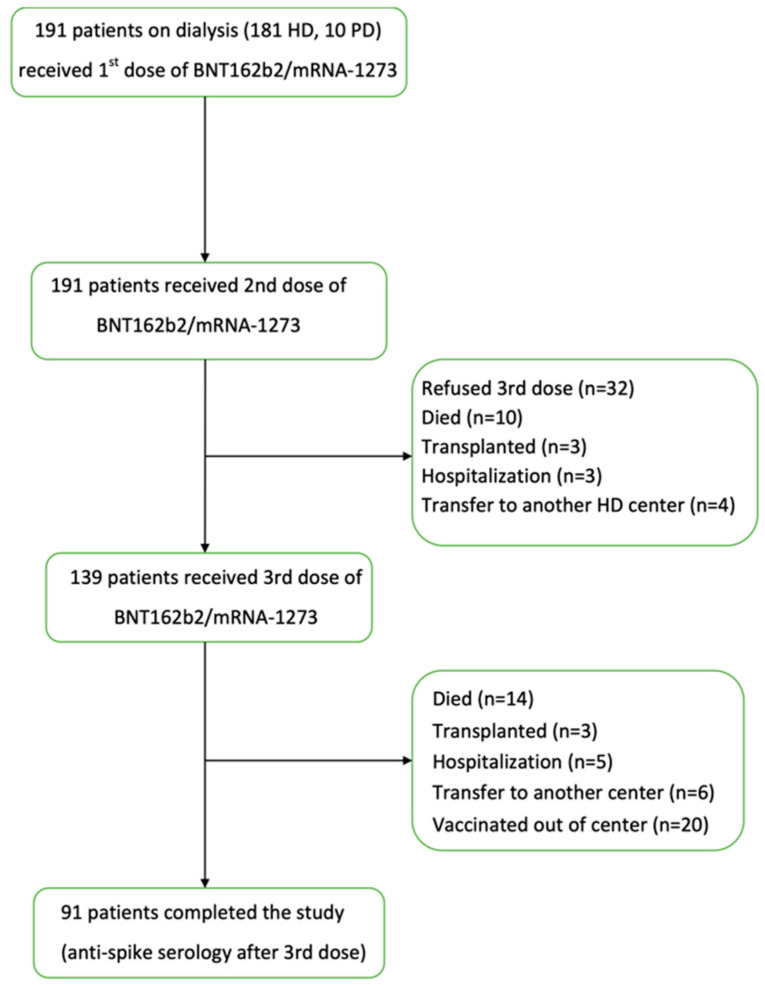
Flow chart of selection of the study population. (HD—hemodialysis, PD—peritoneal dialysis).

**Figure 2 jcm-11-04984-f002:**
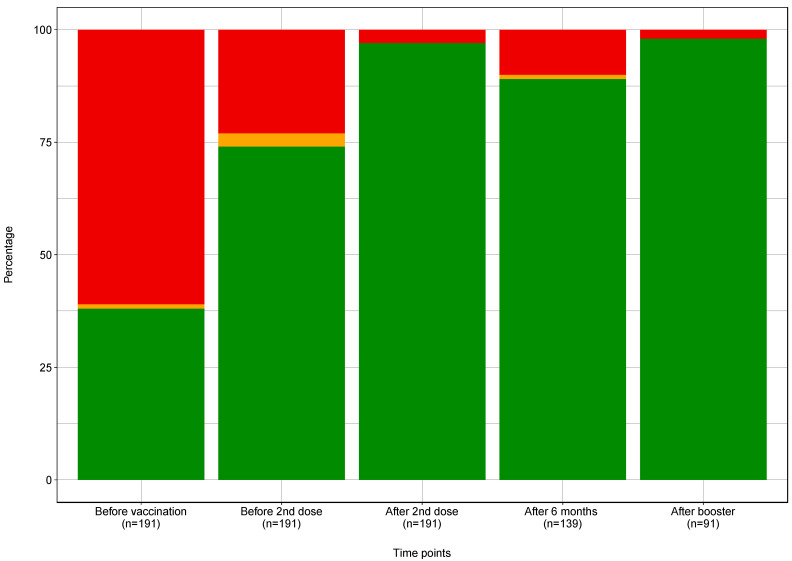
Humoral response after subsequent vaccine doses (red color: non-responders, orange: borderline response, green: adequate response).

**Figure 3 jcm-11-04984-f003:**
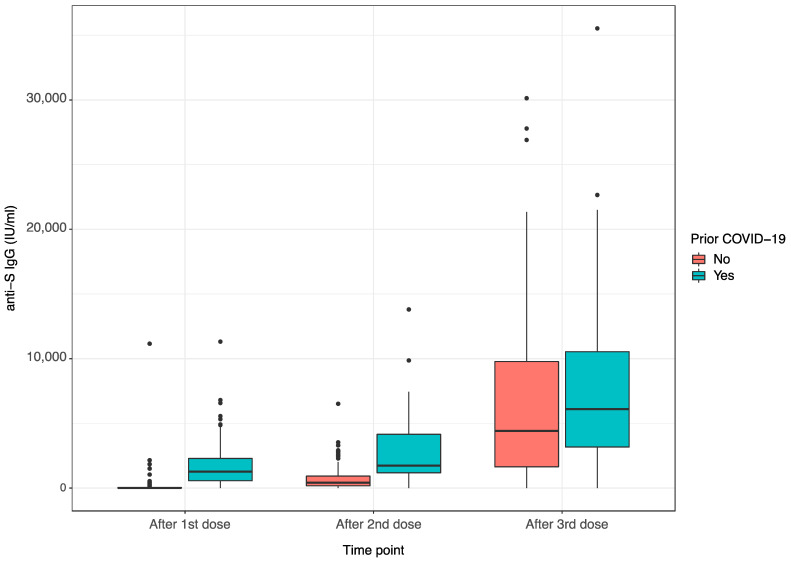
Anti-S IgG levels after subsequent vaccine dose regarding prior COVID-19 infection.

**Figure 4 jcm-11-04984-f004:**
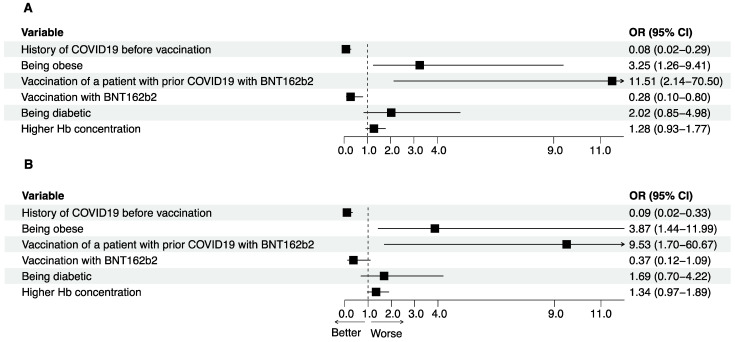
(**A**) Results of the adjusted multivariable analysis of predictive factors of anti-S IgG waning. (**B**) Sensitivity analysis. Results of the adjusted multivariable analysis of predictive factors of anti-S IgG waning after removal of patients who contracted COVID-19 during the study duration. (OR odds ratio, CI confidence interval, black square—odds ratio, black line—95% confidence interval).

**Table 1 jcm-11-04984-t001:** Characteristics of studied population at different time points of study duration.

	A Time Point		
	Baseline (*n* = 191)	After 6 Months (*n* = 139)	After a Booster (*n* = 91)
**Parameter**			
*Demographics*			
Age (yrs. [Q1–Q3])	64.0 [53.0–70.5]	65 [55.0–71.0]	64.0 [56.0–72.0]
Sex (females) *n* (%)	122 (63.9)	83 (59.7)	57 (62.6%)
Dialysis vintage (yrs [Q1–Q3])	2.99 [1.23–4.75]	2.95 [1.25–4.76]	2.82 [1.47–4.85]
*Comorbidities n (%)*			
Obesity	50 (26.2)	40 (28.8)	27 (29.7%)
Diabetes mellitus	59 (30.9)	45 (32.4)	25 (27.5%)
Heart failure	58 (30.4)	47 (33.8)	18 (19.8%)
Hypertension	176 (92.1)	127 (91.4)	81 (89.0%)
Atrial fibrillation	27 (14.1)	24 (17.3)	12 (13.2%)
Prior COVID-19	72 (37.7)	52 (37.4)	32 (35.2%)
*Dialysis modality n (%)*			
Peritoneal dialysis	10 (5.2)	1 (0.7)	1 (1.1 %)
Hemodialysis	181 (94.8)	138 (99.3)	90 (98.9%)
*Type of Vaccine n (%)*			
BNT162b2	109 (57.1)	71 (51.1)	44 (48.5%)
mRNA-1273	82(42.9)	68 (48.9)	47 (51.6%)
*Laboratory, median [Q1–Q3]*			
Hemoglobin (g/dL)	10.7 [9.90–11.5]	10.8 [9.90–11.6]	10.9 [10.2–11.5]
Kt/V	1.40 [1.22–1.60]	1.41 [1.23–1.61]	1.40 [1.23–1.56]

Data are presented as numbers (percentages) or medians [Q1–Q3]. *n* value depicts the number of patients who were evaluated at the specific time point.

**Table 2 jcm-11-04984-t002:** Predictors of adequate humoral response after the 1st, 2nd and 3rd vaccine doses.

**Variable of Interest**	**Independent Variable**	**β**	**SE**	* **p** *
**Adequatehumoral response after** **the first dose**	Age	0.01	0.01	0.41
	Sex (M)	0.30	0.32	0.35
	Obesity	−0.34	0.35	0.34
	Diabetes mellitus	−0.35	0.33	0.30
	Hypertension	0.95	0.66	0.15
	Heart failure	0.33	0.32	0.31
	Atrial fibrillation	0.33	0.42	0.44
	Kt/V	−0.97	0.48	0.04
	Hemoglobin	−0.13	0.12	0.29
	Dialysis modality (PD)	−0.36	0.71	0.61
	Vaccine type (BNT162b2)	1.14	0.32	0.001
**Adequatehumoral response after** **the second dose**	**Independent Variable**	**β**	**SE**	** *p* **
	Age	−0.01	0.03	0.80
	Sex (M)	0.59	0.83	0.49
	Obesity	0.59	1.11	0.60
	Diabetes mellitus	−0.12	0.88	0.90
	Hypertension	0.89	1.13	0.43
	Heart failure	−0.86	0.83	0.30
	Atrial fibrillation	−0.20	1.12	0.86
	Kt/V	−0.88	1.26	0.49
	Hemoglobin	0.03	0.33	0.93
	Dialysis modality (PD)	15.19	2062.64	1.00
	Vaccine type (BNT162b2)	−0.42	0.88	0.63
**Adequatehumoral response after** **the third dose**	**Independent Variable**	**β**	**SE**	** *p* **
	Age	0.09	0.06	0.16
	Sex (M)	−17.25	3040.73	1.00
	Obesity	17.13	3412.21	1.00
	Diabetes mellitus	17.10	3546.07	1.00
	Hypertension	−15.89	3400.72	1.00
	Heart failure	17.00	4179.09	1.00
	Atrial fibrillation	16.00	3104.42	1.00
	Kt/V	1.20	2.42	0.62
	Hemoglobin	0.94	0.56	0.09
	Dialysis modality (PD)	13.78	3956.18	1.00
	Vaccine type (BNT162b2)	−0.07	1.43	0.96

In the case of nominal variables in parentheses, there is a value corresponding to an estimate of the magnitude as well as direction of relation (β). Abbreviations: PD—peritoneal dialysis, M—male, SE—standard error.

**Table 3 jcm-11-04984-t003:** Predictors of anti-S IgG waning. Results of a univariable and multivariable logistic regression (OR odds ratio, CI confidence interval).

Variable of Interest	Predictor	OR (95%CI)
Univariable analysis		
anti-S IgG waning	Demographics	
	Age	1.00 (0.97–1.03)
	Sex	1.03 (0.51–2.06)
	Dialysis vintage	0.97 (0.87–1.08)
	Comorbidities	
	Being obese	3.47 (1.51–8.78) *
	Diabetes mellitus	1.90 (0.90–4.18) *
	Heart failure	1.19 (0.58–2.49)
	Hypertension	1.65 (0.49–5.54)
	Atrial fibrillation	1.07 (0.44–2.74)
	History of COVID-19 infection before vaccination	0.28 (0.13–0.57) *
	*Laboratory*	
	Hemoglobin (g/dL)	1.34 (1.01–1.80) *
	Kt/V	0.89 (0.29–2.63)
	Type of vaccine used (BNT162b2 vs. mRNA-1273)	0.45 (0.22–0.89) *
Multivariable analysis		
anti-S IgG waning	History of COVID-19 infection before vaccination	0.08 (0.02–0.29)
	Being obese	3.25 (1.26–9.41)
	Vaccination of a patient with prior COVID-19 infection with BNT162b2	11.51 (2.14–70.50)
	Vaccination with BNT162b2	0.28 (0.10–0.80)
	Being diabetic	2.02 (0.85–4.98)
	Higher Hb concentration	1.28 (0.93–1.77)

* variables, which predicted anti-IgG waning with probability less than 0.15 and which were included in multivariable model.

**Table 4 jcm-11-04984-t004:** Sensitivity analysis. Results of a multivariable logistic regression after removal of patients who contracted COVID-19 during the study duration. Abbreviations: Hb—hemoglobin.

Variable of Interest	Predictor	OR (95%CI)
anti-S IgG waning	History of COVID-19 infection before vaccination	0.09 (0.02–0.33)
	Being obese	3.87 (1.44–11.99)
	Vaccination of a patient with prior COVID-19 infection with BNT162b2	9.53 (1.70–60.67)
	Vaccination with BNT162b2	0.37 (0.12–1.09)
	Being diabetic	1.69 (0.70–4.22)
	Higher Hb concentration	1.34 (0.97–1.89)

## Data Availability

Detailed data are available on request from the corresponding author.

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
