# Peer review of "Factors Influencing Longevity of Humoral Response to SARS-CoV-2 Vaccination in Patients with End Stage Kidney Disease Receiving Renal Replacement Therapy"

_jcm, 2022, doi:10.3390/jcm11174984_

Round 1

Reviewer 1 Report

Factors influencing longevity of humoral response to SARS-CoV-2 vaccination in patients with end-stage kidney disease receiving renal replacement therapy. by Głowinska I et al

This was an interesting observational paper. The manuscript was generally well written. While I appreciate the authors’ first language is not English the manuscript would benefit from further proof-reading to correct a few minor grammatical errors. Eg Authors use of ‘since’ when they mean ‘after’

It was not clear whether the patients received 3 doses of the same vaccine or whether the booster vaccine was changed to the other of the two vaccines offered – could the authors please comment.

Fig 1  

Table 1 The column labelled ‘value’ contains both continuous variables and patient numbers with percentages. For clarity perhaps the authors could consider splitting the table into two parts and labelling the columns appropriately.

Fig 2. This table would benefit from an ‘n’ number under each column. 72 patients were seropositive before the start of the study but only 91 finished. I appreciate percentages are stated but it would help to know the numbers of patients at each stage.     

Fig 3. Out of interest how many of the seropositive patients prior to vaccination had had a ‘positive test’ (by PCR or lateral flow)?  Were any of these patients asymptomatic for COVID? Did any patients contract COVID during the study? Had any of the patients who were not seropositive before vaccination, had prior COVID infection but their immunity had waned?

It would be interesting to know the calendar time frame of the study to have an idea within which  phase of the pandemic it took place, giving a better idea of the potential exposure time of the dialysis population to the circulating virus and also the prevailing virus type eg WT, Alpha, Delta, Omicron? The humoral response of dialysis patients to the various types is known to be different.

Reviewer 2 Report

The authors describe in a real world setting the response of dialysis patients to COVID 19 mRNA vaccinations from pre-first dose Ab levels to post booster levels.  They highlight the significant decrease in Ab levels in obese patients.  They show similar findings in regards to increased Ab response in people with prior infection re-enforcing their data findings.

Line 72: 2.2: Can you show describe the time range of the booster dose which was given 6 months after 2nd dose? Did the relative time between the second dose and the booster dose have an effect on antibody production?

Line 103:  Would it be more clear to add:  ... concordant with seroconversion and therefore were considered to have had a prior COVID19 infection.  

Line 108:  Were any deaths or hospitalizations due to COVID19 infection?  Do you have any clinical information on symptomatic infections in this group?  Your later statements suggest you may not, but if you do, it would be helpful.

Line 122: do you have any additional information concerning the two patients who did not respond to the booster dose?  Did they have a particular risk factor such as have an immunodeficiency disease or being on immunosuppression or recent chemotherapy or the like?

Line 173: reword to “To our best knowledge,” …

A figure showing the decline in antibody levels in obese patients for each of the two vaccination types would be of interest.
